# Investigation of the Phase Transition Mechanism in LiFePO_4_ Cathode Using In Situ Raman Spectroscopy and 2D Correlation Spectroscopy during Initial Cycle

**DOI:** 10.3390/molecules24020291

**Published:** 2019-01-14

**Authors:** Yeonju Park, Soo Min Kim, Sila Jin, Sung Man Lee, Isao Noda, Young Mee Jung

**Affiliations:** 1Department of Chemistry, Institute for Molecular Science and Fusion Technology, Kangwon National University, Chuncheon 24341, Korea; yeonju4453@kangwon.ac.kr (Y.P.); italy3255@naver.com (S.M.K.); jsira@kangwon.ac.kr (S.J.); 2Department of Nano Applied Engineering, Kangwon National University, Chuncheon 24341, Korea; smlee@kangwon.ac.kr; 3Department of Materials Science and Engineering, University of Delaware, Newark, DE 19716, USA

**Keywords:** Li-ion Battery, LiFePO_4_, two-dimensional correlation spectroscopy, in situ Raman spectroscopy

## Abstract

The phase transition of the LiFePO_4_ and FePO4 in Li-ion cell during charging-discharging processes in the first and second cycles is elucidated by Raman spectroscopy in real time. In situ Raman spectroscopy showed the sudden phase transition between LiFePO_4_ and FePO4. Principal component analysis (PCA) results also indicated that the structural changes and electrochemical performance (charge-discharge curve) are correlated with each other. Phase transition between LiFePO_4_ and FePO4 principally appeared in the second charging process compared with that in the first charging process. 2D correlation spectra provided the phase transition mechanism of LiFePO_4_ cathode which occurred during the charging-discharging processes in the first and second cycles. PCA and 2D correlation spectroscopy are very helpful methods to understand in situ Raman spectra for the Li-ion battery.

## 1. Introduction

Battery performance, duration, and stability can be determined during the initial cycle. Therefore, it is important to understand structural changes in cathode and anode materials, formation of solid electrolyte interphase (SEI) film, electrical performance, etc. in real time. In the past few years, many researchers have tried to monitor in real time reactions inside batteries during electrochemical performance by using XRD [1,2,3,4], XAS [5,6,7], Raman [8,9,10,11,12,13,14,15,16,17], NMR [18], and so forth [19,20]. However, the studies for electrochemical reactions during the initial cycle is still limited.

LiFePO_4_ is one of the cathode materials having lower cost, environmental compatibility, high theoretical specific capacity of 170 mAhg^−1^ and especially a superior safety performance [21,22]. Therefore, it is spotlighted for large size lithium-ion (Li-ion) batteries, such as the energy sources in hybrid-electric vehicles (HEVs) or electric vehicles (EVs), and energy grid storage. Nevertheless, application of LiFePO_4_ cathode material is limited owing to low electronic conductivity and poor Li-ion diffusion [23]. To overcome these problems, LiFePO_4_ cathode materials have been alternated using carbon coating [13,22] or extending the surface area [23].

LiFePO_4_ cathode material has flat charging and discharging profiles at ~3.4 V vs. Li/Li^+^ resulting from LiFePO_4_ phase transition [24]_._ Many researchers studied the phase transition of LiFePO_4_ structure during charging-discharging processes. T. Yamanaka et al. [25] and other groups [26,27,28] reported that LiFePO_4_ separates to LiFePO_4_ and FePO_4_ phases during the charging process, having a miscibility gap at the Li concentration between them. The so-called ‘core shrinking’ theoretical model of the phase transition has been described by A.S. Andersson et al. and V. Srinivasan et al. [29,30]. Recently, C. Delmas et al. reported another phase transition mechanism, which is named the ‘domino-cascade’ model [31]. They assume that phase transformations at particle-level are fast, and small particles appear either fully charged (FePO_4_ phase) or discharged (LiFePO_4_ phase) due to a very rapid two-phase front [4,31]. The phase transition mechanism of LiFePO_4_ has been also investigated through in situ Raman spectroscopy [4,13]. Many models of phase transformation in LiFePO_4_ were reported; however the understanding is still limited. The interpretation of the highly overlapped spectra obtained during the charging–discharging processes remains difficult because the electrochemical reactions in the batteries are very complicated [32].

Nowadays generalized two-dimensional correlation spectroscopy (2D-COS) is accepted as one of the tools applicable to the in-depth analysis of various spectral data. 2D-COS can sort out important information in systematic variations of almost any reasonable analytical signals observed with a variety of probes under various forms of applied external perturbations. The perturbation-induced spectral variation of systems can be analyzed by a form of cross correlation analysis in 2D-COS. This approach has been applied to better understand complex systems, such as Li-ion batteries, because 2D-COS provides better information that is not readily detected with conventional spectroscopic measurements. In our previous study, electrochemical reactions of electrode materials using 2D-COS have successfully elucidated [33,34,35,36,37,38,39,40,41] analysis of in situ Raman spectra of LiCoO_2_/Li cell [32].

Principal component analysis (PCA) is a very popular analytical technique for spectroscopy. It can decompose the original data into the products of scores and loading vectors and provides a precise mathematical estimation of changes along sample and variable vectors. From PCA results, we can capture the dynamics of systems in response to perturbations and to display relationships among variables or among variables and systems [42,43,44].

In this study, PCA and 2D-COS were applied to in situ Raman spectroscopy for deeper understanding of phase transition of LiFePO_4_ cathode. PCA results adequately explain the correlation of spectral changes and electrochemical performance. We also deduced the phase transition mechanism during the first and second charging-discharging processes using 2D correlation spectroscopy.

## 2. Results and Discussion

We measured in situ Raman spectra during charging-discharging processes in the first and second cycles. Figure 1A shows charging and discharging profiles of the first and second cycles of a Li-ion cell with LiFePO_4_ cathode. There are plates region at 3.48 and 3.38 V and 3.44 and 3.40 V during the first and second charging-discharging processes, respectively. This is the most noticeable feature of LiFePO_4_ material [24]. The sharp changes were observed in the other potential near the end of charging and near the end of discharging in the first and second cycles. From this result, we predict that lattices structure, Li ion diffusion rate, phase transition suddenly changed near the end of charging and discharging in the first and second cycles. In situ Raman spectra were collected during the first charging-discharging processes every 1 h, especially those that were recorded every 5 min and 2 min during near the end of charge in the first and second cycles, respectively, as shown in Figure 1B.

Unexpected spectral changes were detected near the end of charging in the first (10 h) and second cycles (9 h), respectively. The bands at 960 and 1020–1130 cm^−1^, assigned to symmetric and asymmetric stretching of PO_4_^3−^ in FePO_4_ phase [13], suddenly appeared at the end of charging then suddenly disappeared after charging. On the other hand, a band at 950 cm^−1^, corresponds to the symmetric stretching of PO_4_^3−^ in LiFePO_4_ phase, abruptly fades at the end of charging then abruptly came out after charging. As shown in in situ Raman spectra recorded near the end of charging of the first and second cycles, the variations of the bands at 950, 960 and 1020–1130 cm^−1^ also suddenly occurred, which were not gradual changes. This observation means that phase transition happened promptly near the end of charge. We also observed that the intensity of these bands decreased during the first and second charging-discharging processes. This result indicates that optical skin depth for the LiFePO_4_ cathode may change without any thermal damages because of laser exposure. If the electrode is harmed by laser, a new band at 1003 cm^−1^ assigned to a disordered phase of FePO_4_ should have appeared [45].

Because the spectral region of PO_4_^3−^ stretching between 930 and 980 cm^−1^ is only presented as the existence of both LiFePO_4_ and FePO_4_ [13], we only focus this region as shown in Figure 2. In Figure 2, phase transition between LiFePO_4_ and FePO_4_ phases were obviously monitored. During the first and second cycles, the band at 950 cm^−1^, observed in open circuit voltage (OCV), changed to 960 cm^−1^ at the end of charging and then repositioned at 950 cm^−1^ after charging. The band shift from 950 to 948 cm^−1^ was observed only in the end of discharging of the first cycle, not in that of the second cycle. This difference clearly indicates that the more stable structure of LiFePO_4_ may be formed after the first cycle. Therefore, we examined when the phase changes happened during the second charging process. Through Appendix A, we speculate that the potential range that appeared during phase transition is between 3.6 and 3.7 V during the second charging process.

To understand the correlation between the structural changes and the performances of the Li-ion cell, we also carried out PCA for the in situ Raman spectra of LiFePO_4_/Li cell during charging-discharging processes in the first and second cycles. We applied PCA to in situ Raman spectra obtained from Figure 1. Appendix A presented plots of the first four scores, the loading values, and the loading vectors in 930–980 cm^−1^ region, respectively, of the in situ Raman spectra for LiFePO_4_ cathode in Li-ion cell during the charging-discharging processes in the first and second cycles. Interestingly, the variation trends of score plots for the first principal component (PC 1) are in good agreement with those of CV curve of LiFePO_4_/Li cell during the charging-discharging processes in the first and second cycles, as shown in Appendix A. As shown in loading vectors of Appendix A, the first four principal components (PCs) of the bands at 948 and 951 cm^−1^ have positive, positive, negative, and positive intensities, respectively, indicating that LiFePO_4_ phase mainly changed at 1 h between 8 h 40 min in the charging process of the second cycle. In contrast, the band at 960 cm^−1^ has negative, positive, positive, and positive intensities in the first four PCs, respectively, indicating that FePO_4_ phase principally varied at 8 h 50 min and 8 h 52 min in charging process of the second cycle. PCA results clearly identify the LiFePO_4_ and FePO_4_ phases. The phase transition between LiFePO_4_ and FePO_4_ occurred even during the first charging process, but they are not clearly detected in PCA results. This result shows that phase transition of LiFePO_4_ and FePO_4_ in the first charging process was not as strong as the second charging process.

To deeper understand the structural changes in LiFePO_4_ cathode in Li-ion cell during the charging-discharging processes, 2D-COS was applied to in situ Raman spectra of LiFePO_4_/Li cell during the charging-discharging processes in the first and second cycles. Figure 3 and Figure 4 displayed 2D correlation in situ Raman spectra during the charging-discharging processes in the first and second cycles.

We analyzed firstly in situ Raman spectra of LiFePO_4_/Li cell during the charging-discharging processes of the first cycle. As shown in Figure 3, 2D correlation spectra of the charging-discharging processes of the first and second cycles were different. During the first charging process, two auto peaks at 946 and 960 cm^−1^ were observed in the synchronous 2D correlation spectrum as shown in Figure 3A. Three positive cross peaks at the (950,946), (958,948), and (961,958) cm^−1^ were observed in the asynchronous 2D correlation spectrum. They have positive, negative, and positive cross peaks, respectively, in the corresponding synchronous 2D correlation spectrum. From the sign of the cross peaks in the synchronous and asynchronous 2D correlation spectra, we determine the sequential order of the band intensity changes with time as follows; 950 (due to LiFePO_4_ phase) → 946 (due to LiFePO_4_ phase) → 958 (due o FePO_4_ phase) → 961 (due to FePO_4_ phase) cm^−1^. This result means that LiFePO_4_ phase distorted, then FePO_4_ phase appeared, and finally FePO_4_ phase distorted during the first charging process. During the first discharging process, one auto peak at 945 cm^−1^ is detected in the synchronous 2D correlation spectrum as shown in Figure 3C. In the asynchronous spectrum (Figure 3D), a lot of cross peaks appeared associated with noise. This observation means that the intensity of a band corresponded distorted LiFePO_4_ phase only changed in the first discharging process. In other words, phase transition in LiFePO_4_ structure did not happen during the first discharging process.

2D correlation spectra during the charging-discharging processes in the second cycle are displayed as shown in Figure 4. In the second charging process, two cross peaks at (951,946) and (960,951) cm^−1^ appeared in the asynchronous spectrum (Figure 4B). Based on the sign of the cross peaks in 2D correlation spectra, we deduced the following sequence of the intensity variations: 960 (due to FePO_4_ phase) → 951 (due to LiFePO_4_ phase) → 946 (due to LiFePO_4_ phase). Variation of FePO_4_ phase occurred before changes of LiFePO_4_ phase and finally LiFePO_4_ phase distorted. Interestingly, FePO_4_ phase changes appeared in advance unlike the first charging process. This indicates that the FePO_4_ phase does not appear abruptly in near the end of charging during the second cycle. This observation means that variations of the FePO_4_ phase were formed during the first cycle taking place before the LiFePO_4_ phase change. During the second discharging process, two cross peaks at (953,948) and (960,953) cm^−1^ appeared in the asynchronous spectrum (Figure 4D). A band at 960 cm^−1^ is not detectable in 1D in situ Raman spectra during the second discharging process. These results indicate that the FePO_4_ phase existed in a small of amount but not disappeared suddenly. From the 2D correlation analysis, the sequential order of the band intensity changes can be determined as follows: 948 (due to LiFePO_4_ phase) → 953 (due to LiFePO_4_ phase) → 960 (due to FePO_4_ phase) cm^−1^. The event of the band intensity changes during the second discharging process represented exactly opposite trend with that during the second charging process. In other words, reversible phase transition was shown after the first cycle. From this result we conclude that LiFePO_4_ structure is getting stable after the first cycle. As shown in Appendix A, the power spectra, which were extracted along the diagonal line on the synchronous 2D correlation spectra of Figure 3A,C, and Figure 4A,B, indicate that the phase transition between LiFePO_4_ and FePO_4_ phases appeared more sensitively during the second charging process. This result is in a good agreement with the PCA result.

## 3. Materials and Methods

The electrochemical behavior of LiFePO_4_ was investigated using a modified coin cell (2032 type). The modified coin cell was fabricated using the method described by Wu et al. [13]. In this study, a 1 mm hole was drilled in the back of a 2032 type coin cell (Wellcos Corporation, Gunpo, Korea), and the cell was used as the cathode can. A slurry consisting of 90 wt % LiFePO_4_ (Nippon Chemical Co., Tokyo, Japan) powder, 5 wt % Super P (conductor), and 5 wt % polyvinylidene fluoride (PVdF, binder) dissolved in 1-methyl-2-pyrrolidinone (NMP) was prepared. Cathodes were made by coating the slurry (thickness was 200 μm) onto a mesh-type aluminum foil substrate (Dexmet Corporation, Wallingford, CT, USA). Using these electrodes, the test cells were fabricated with metallic Li anodes and two separators (Celgard 2400, Celgard Co., NC, USA and Amotech Co., Incheon, South Korea) in an Ar-filled glove box. The electrolyte was a 1.0 M solution of LiPF_6_ in EC: DEC = 1:1 by volume (PANAX ETEC Co., Busan, Korea). To evaluate the cell performance, the test cells were aged for 4 h at 40 °C in a vacuum oven. Then, the first and second charge-discharge cycles were galvanostatically performed at rates of 0.1 C, at a constant current of 3.0–4.2 V at room temperature using a CH660E electrochemical analyzer (CH Instruments Inc., Bee Cave, TX, USA).

The in situ Raman spectra were obtained at room temperature during the first and second charging-discharging processes using a Jobin Yvon/HORIBA HR evolution Raman spectrometer equipped with an integral BX 41 confocal microscope. Radiation from an air-cooled frequency-doubled Nd:YAG laser (532 nm) was used as the excitation source. Raman scattering was detected at a 180° geometry using a multichannel air-cooled (−60 °C) charge-coupled device (CCD) camera (1024 × 256 pixels). We performed Raman shift calibration with Si wafer, which has a characteristic Raman shift at 520 cm^−1^. The in situ Raman spectra were collected with 4 s exposure time and 10 accumulations using 50× (NA 0.8) objective during the charging-discharging processes in the first and second cycles. The power and spot size of the laser were about 12.5 mW and 8 μm, respectively. All spectra were baseline corrected, smoothed and normalized using the PLS_Toolbox 8.1 software (Eigenvector Research, Inc., Wenatchee, WA, USA) for MATMAB R2017b (The Mathworks Inc., Natick, MA, USA). Smoothing was performed by Saviski-Golay method with 3 points filter width. Area normalization carried out in the 1150–1800 cm^−1^ region corresponding to carbon of conductor (Super P) because we only focused on the spectral variation in 930–980 cm^−1^ in this study. We also performed PCA analysis with PLS_Toolbox 8.1 software (Eigenvector Research, Inc., Wenatchee, WA, USA) for MATMAB R2017b (The Mathworks Inc., Natick, MA, USA). Synchronous and asynchronous 2D correlation in situ Raman spectra were obtained using MATLAB. The red and blue lines represent positive and negative cross peaks, respectively.

## 4. Conclusions

In situ Raman spectroscopy is a useful method to monitor the phase transition of LiFePO_4_ cathode during the first and second charging-discharging processes. By applying the PCA and 2D-COS to in situ Raman spectra, we can get more information, which was not easily detected using only 1D spectra. As PC 1 score plots are in good agreement with the trend of charging and discharging profiles, spectral changes affect electrochemical performance. The phase transition mechanism of the LiFePO_4_ cathode in LiFePO_4_/Li ion cell can be examined using 2D correlation spectra obtained from in situ Raman spectra during the first and second charging-discharging processes.

## Figures and Tables

**Figure 1 molecules-24-00291-f001:**
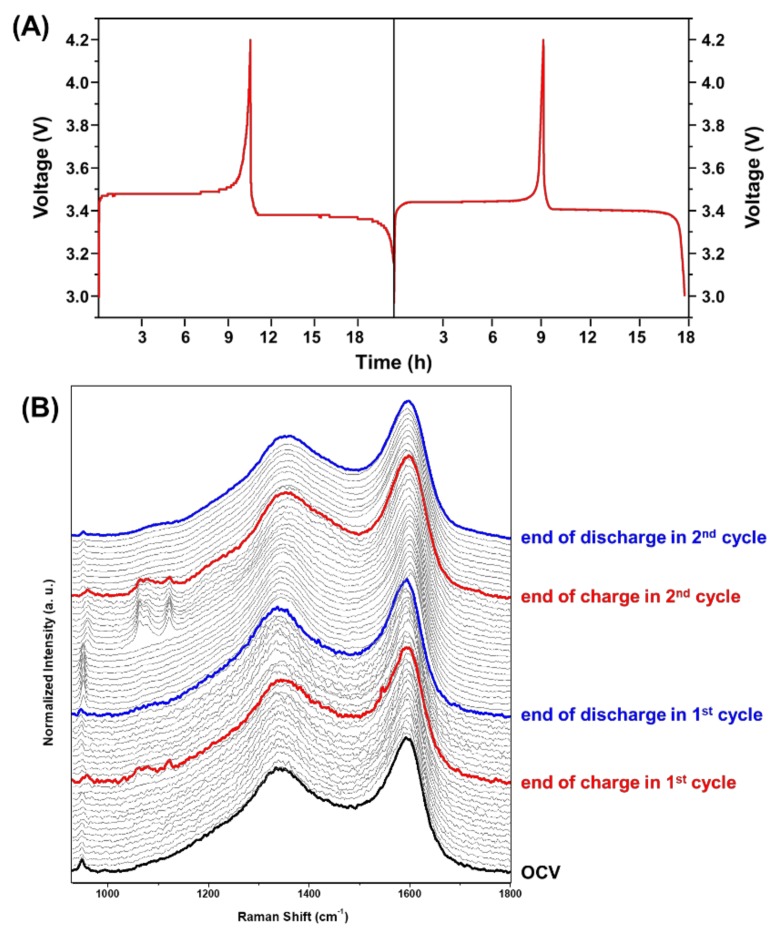
(**A**) Charging and discharging profiles during the first and second cycles of a LiFePO_4_/Li cell. (**B**) In situ Raman spectra of LiFePO_4_ cathode in the 930–1800 cm^−1^ region measured during the charging-discharging processes of the first and second cycles of LiFePO_4_/Li cell.

**Figure 2 molecules-24-00291-f002:**
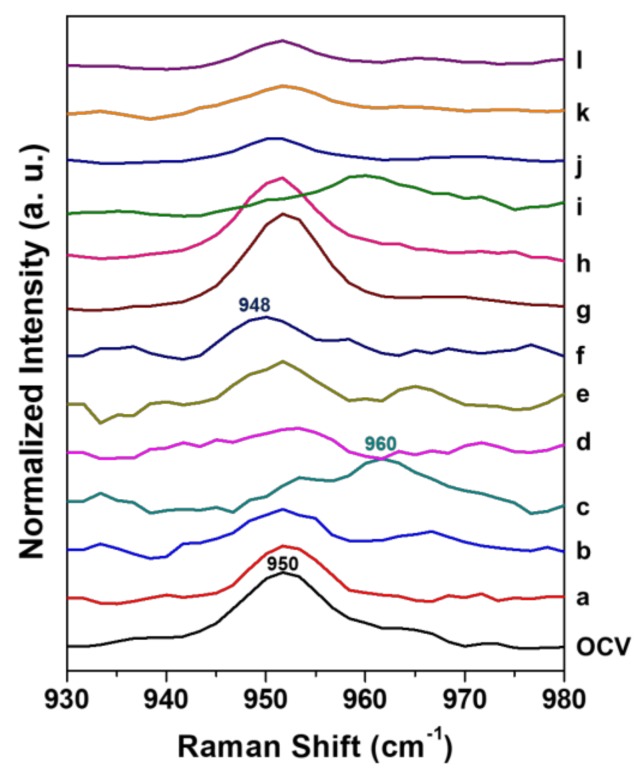
In situ Raman spectra in the PO_4_^3−^ stretching region (930–980 cm^−1^) of OCV, a–f: the first cycle (a: 1 h, b: 7 h, c: 10 h 30 min (end of charge), d: 11 h, e: 18 h, f: 20 h (end of discharge)) and g–i: the second cycle (g: 1 h, h: 7 h, i: 9 h 8 min (end of charge), j: 10 h, k: 15 h, l: 18 h (end of discharge)).

**Figure 3 molecules-24-00291-f003:**
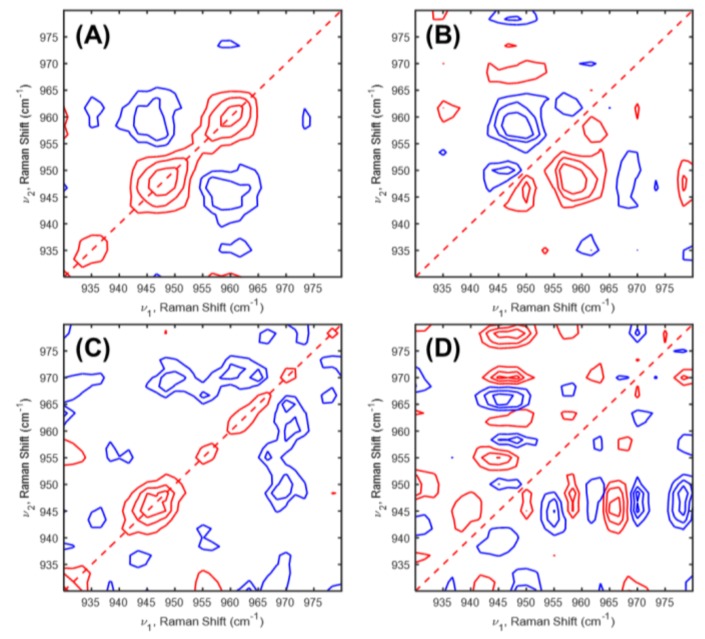
(**A**,**C**) Synchronous and (**B**,**D**) asynchronous 2D correlation spectra of in situ Raman spectra during the (**A**,**B**) charging and (**C**,**D**) discharging processes in the first cycle obtained Figure 1.

**Figure 4 molecules-24-00291-f004:**
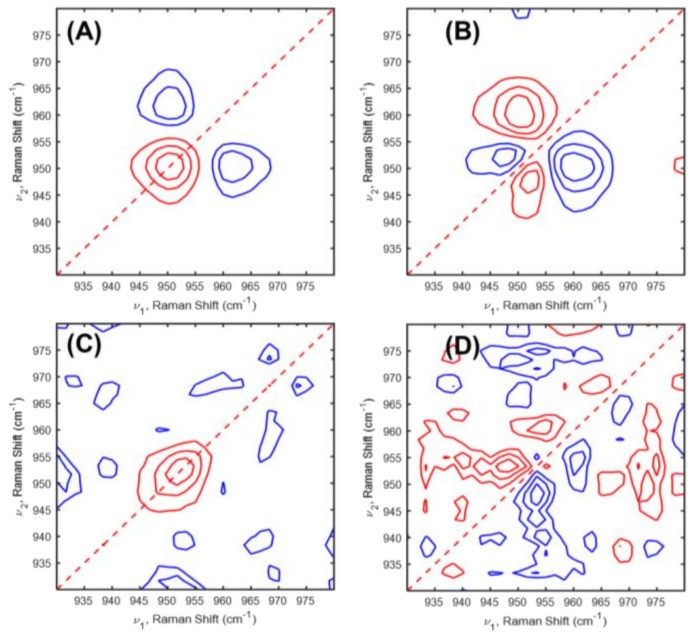
(**A**,**C**) Synchronous and (**B**,**D**) asynchronous 2D correlation spectra of in situ Raman spectra during the (**A**,**B**) charging and (**C**,**D**) discharging processes in the second cycle obtained Figure 1.

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
