# Peer review of "Investigation of the Phase Transition Mechanism in LiFePO4 Cathode Using In Situ Raman Spectroscopy and 2D Correlation Spectroscopy during Initial Cycle"

_molecules, 2019, doi:10.3390/molecules24020291_

Round 1
Reviewer 1 Report
The authors presented the use of Raman spectroscopy and PCA analysis to study in situ the charging-discharging processes in Li-ion batteries.
The work presented by the authors is interesting. English quality is to be improved as at the moment it is not always easy to fully understand the content of the manuscript. The manuscript must also be checked thoroughly for spelling and grammar mistakes. Moreover, a large part of the discussion is based on figures and tables presented as supplemental material. This makes it difficult to follow the discussion of the results. The relevant figures and tables should be present in the article.
Before the manuscript can be accepted for publication, the following points should be addressed.
Introduction
Line 39: Sentence “LiFePO4 cathode material has limited owing to” is incomplete. Check English.
Lines 48-49: Sentence “Recently, another phase transition mechanism reported by C. Delmas et al., which is named ‘domino-cascade’ model” is incomplete. Check English
Line 55: A brief explanation of 2D-COS technique is needed.
Line 53: Clarify the sentence “some methods may require …. before bacteria identification”. Clarify what these methods are
Results and discussion
Line 71: Authors should clarify what “other potential” they are referring to
Figure 1: Caption is confusing as it is not clear what description belongs to what. Perhaps it would be advisable to have the letter (A), (B) and (C) in front of the description rather than after. Authors should explain the meaning of the labels “c” and “i” that appear in figure 1A.
Line 72: How did the authors predict the changes in lattice structure, Li ion diffusion rate and phase transition? Explain further
Line 73-75: Sentence “In situ Raman ….. as shown in Figure S1” is unclear. Check English
Lines 76-88: English needs to be improved as well as style (at the moment there are a series of very short and not always correlated sentences) to make the content clearer. In addition, there are references to Raman peaks that are not shown in the spectra in Fig.1. The relevant spectra should be shown.
Line 99: The PCA results shown in Fig. 1 are not discussed. Moreover, the discussion of the PCA findings refer to figures which are presented as supplemental material. For clarity’s sake it would be better if these figures were shown in the article.
Line 105: More explanation is required about what information PC1 is carrying and what the apparent agreement with the CV curve means and implies
Lines 130-132: Sentence “During the ….. Figure 2(A)” is incomplete. Check English
Lines 132-134: Sentences “Three positive cross peaks ….. 2D correlation spectrum” are unclear. Check English
Line 154: Sentence “This indicates ………. end of charge” is unclear. Check English
Materials and Methods
Line 189: What were the specifications of the objective used for Raman spectroscopy experiments? What was the power of the laser and how much power was used on the sample?
Line 200: Sentence “Area normalisation … to conductor (Super P)” is unclear. Check English. Authors should explain why the area of the spectra were normalised to the area of the 1150 – 45000px-1 region
PCA analysis should be described
Conclusions
Line 207: Sentence “PC 1 score plot is coincident with the charging and discharging profiles, therefore, spectral changes affect to electrochemical performance” is unclear. Check English
Author Response
All the replied are written in boldface.
We appreciate reviewer’s comments on our manuscript which have made us to revise the manuscript significantly.
Response on comments of Reviewers #1
The authors presented the use of Raman spectroscopy and PCA analysis to study in situ the charging-discharging processes in Li-ion batteries.
The work presented by the authors is interesting. English quality is to be improved as at the moment it is not always easy to fully understand the content of the manuscript. The manuscript must also be checked thoroughly for spelling and grammar mistakes. Moreover, a large part of the discussion is based on figures and tables presented as supplemental material. This makes it difficult to follow the discussion of the results. The relevant figures and tables should be present in the article.
Before the manuscript can be accepted for publication, the following points should be addressed.
Introduction
Line 39: Sentence “LiFePO4 cathode material has limited owing to” is incomplete. Check English.
According to Reviewer’s comments, we revised this sentence in the revised manuscript.
“Nevertheless, application of LiFePO4 cathode materials is limited owing to low electronic conductivity and poor Li-ion diffusion[23].”
This content appears at the Introduction (in page 1) of the revised manuscript.
Lines 48-49: Sentence “Recently, another phase transition mechanism reported by C. Delmas et al., which is named ‘domino-cascade’ model” is incomplete. Check English
According to Reviewer’s comments, we revised this sentence in the revised manuscript.
“Recently, C. Delmas et al. reported another phase transition mechanism, which is named ‘domino-cascade’ model[31].”
This content appears at the Introduction (in page 2) of the revised manuscript.
Line 55: A brief explanation of 2D-COS technique is needed.
According to Reviewer’s comments, we added the brief explanation of 2D-COS in the revised manuscript.
“2D-COS can sort out important information in systematic variations of almost any reasonable analytical signals observed with a variety of probes under various forms of applied external perturbations. The perturbation-induced spectral variation of systems can be analyzed by a form of cross correlation analysis in 2D-COS.”
This content appears at the Introduction (in page 2) of the revised manuscript.
Line 53: Clarify the sentence “some methods may require …. before bacteria identification”. Clarify what these methods are
We did not mention that sentence in the manuscript.
Results and discussion
Line 71: Authors should clarify what “other potential” they are referring to
According to Reviewer’s comments, we revised the sentence in the revised manuscript.
“The sharp changes were observed in the other potential near the end of charging and near the end of discharging in the first and second cycles.”
This content appears at the Results and Discussion (in page 2) of the revised manuscript.
Figure 1: Caption is confusing as it is not clear what description belongs to what. Perhaps it would be advisable to have the letter (A), (B) and (C) in front of the description rather than after. Authors should explain the meaning of the labels “c” and “i” that appear in figure 1A.
According to Reviewer’s comments, we corrected the Figure caption in the revised manuscript.
“Figure 2. Charging and discharging profiles during the first and second cycles of a LiFePO4/Li cell. (B) In situ Raman spectra in the PO43- stretching region (930-980 cm-1) of OCV, a-f: the first cycle (a: 1 h, b: 7 h ,c: 10 h 30 min (end of charge), d: 11 h, e: 18 h ,f: 20 h (end of discharge)) and g-i: the second cycle (g: 1 h, h: 7 h , i: 9 h 8 min (end of charge), j: 10 h, k: 15 h ,l: 18 h (end of discharge)).”
This content appears at the Results and Discussion (in page 3) of the revised manuscript.
We already explained the meaning of the labels “c” and “i” in the manuscript.
Line 72: How did the authors predict the changes in lattice structure, Li ion diffusion rate and phase transition? Explain further
It is well known that the change of the charging-discharging curve is affected by structural changes, SEI formation, increased resistance, and so one. Therefore, we can predict such changes in this study.
Line 73-75: Sentence “In situ Raman ….. as shown in Figure S1” is unclear. Check English
According to Reviewer’s comments, we revised the sentences in the revised manuscript.
“In situ Raman spectra were collected during the first charging-discharging processes every 1 h, especially that were recorded every 5 min and 2 min during near the end of charge in the first and second cycles, respectively, as shown in Figure 1(B).”
This content appears at the Results and Discussion (in page 2) of the revised manuscript.
Lines 76-88: English needs to be improved as well as style (at the moment there are a series of very short and not always correlated sentences) to make the content clearer. In addition, there are references to Raman peaks that are not shown in the spectra in Fig.1. The relevant spectra should be shown.
According to Reviewer’s comments, we corrected the sentences and added the reference [13] in the revised manuscript.
“Unexpected spectral changes were detected near the end of charging in the first (10 h) and second cycles (9 h), respectively. The bands at 960 and 1020-1130 cm-1, assigned to symmetric and asymmetric stretching of PO43- in FePO4 phase[13], suddenly appeared at the end of charging then suddenly disappeared after charging. On the other hand, a band at 950 cm-1, correspond to symmetric stretching of PO43- in LiFePO4 phase, abruptly fades at the end of charging then abruptly came out after charging. As shown in in situ Raman spectra recorded near the end of charging of the first and second cycles, the variations of the bands at 950, 960 and 1020-1130 cm-1 also suddenly occurred, which were not gradual changes.”
This content appears at the Results and Discussion (in page 2) of the revised manuscript.
Line 99: The PCA results shown in Fig. 1 are not discussed. Moreover, the discussion of the PCA findings refer to figures which are presented as supplemental material. For clarity’s sake it would be better if these figures were shown in the article.
We already described PCA results shown in Fig. S1 and Fig. S2 at line 117 to 128 of the manuscript.
Line 105: More explanation is required about what information PC1 is carrying and what the apparent agreement with the CV curve means and implies
According to Reviewer’s comments, we corrected the sentence in the revised manuscript.
“Interestingly, the variation trends of score plots for the first principal component (PC 1) are in good agreement with those of CV curve of LiFePO4/Li cell during the charging-discharging processes in the first and second cycles, as shown in Figure 2S.”
This content appears at the Results and Discussion (in page 3) of the revised manuscript.
Lines 130-132: Sentence “During the ….. Figure 2(A)” is incomplete. Check English
According to Reviewer’s comments, we revised the sentence in the revised manuscript.
“During the first charging process, two autopeaks at 946 and 960 cm-1 were observed in the synchronous 2D correlation spectrum as shown in Figure 3(A).”
This content appears at the Results and Discussion (in page 4) of the revised manuscript.
Lines 132-134: Sentences “Three positive cross peaks ….. 2D correlation spectrum” are unclear. Check English
According to Reviewer’s comments, we revised the sentence in the revised manuscript.
“Three positive cross peaks at the (950,946), (958,948), and (961,958) cm-1 were observed in the asynchronous 2D correlation spectrum. They have positive, negative, and positive cross peaks, respectively, in the corresponding synchronous 2D correlation spectrum.”
This content appears at the Results and Discussion (in page 4) of the revised manuscript.
Line 154: Sentence “This indicates ………. end of charge” is unclear. Check English
According to good Reviewer’s comments, we revised the sentence in the revised manuscript.
“This indicates that FePO4 phase does not appeared abruptly in near the end of charging during the second cycle.”
This content appears at the Results and Discussion (in page 5) of the revised manuscript.
Materials and Methods
Line 189: What were the specifications of the objective used for Raman spectroscopy experiments? What was the power of the laser and how much power was used on the sample?
According to Reviewer’s comments, we added the objective specifications and laser power in the revised manuscript.
“The in situ Raman spectra were collected with 4 s exposure time and 10 accumulations using 50x (NA 0.8) objective during the charging-discharging processes in the first and second cycles. The power and spot size of the laser were about 12.5 mW and 8 mm, respectively.”
This content appears at the Materials and Methods (in page 6) of the revised manuscript.
Line 200: Sentence “Area normalisation … to conductor (Super P)” is unclear. Check English. Authors should explain why the area of the spectra were normalised to the area of the 1150 – 45000px-1 region
According to Reviewer’s comments, we revised the sentence in the revised manuscript.
“Area normalization carried out in 1150-1800 cm-1 region corresponding to carbon of conductor (Super P) because we only focused the spectral variation in 930-980 cm-1 in this study.”
This content appears at the Materials and Methods (in page 6) of the revised manuscript.
PCA analysis should be described
According to Reviewer’s comments, we revised the sentence in the revised manuscript.
“We also performed PCA analysis with PLS_Toolbox 8.1 software (Eigenvector Research, Inc., Wenatchee, WA, USA) for MATMAB R2017b (The Mathworks Inc., Natick, MA, USA).”
This content appears at the Materials and Methods (in page 6) of the revised manuscript.
Conclusions
Line 207: Sentence “PC 1 score plot is coincident with the charging and discharging profiles, therefore, spectral changes affect to electrochemical performance” is unclear. Check English
According to Reviewer’s comments, we corrected the sentence as follows
“As PC 1 score plot in good agreement with the trend of charging and discharging profiles, spectral changes affect to electrochemical performance.”
This content appears at the Conclusions (in page 6) of the revised manuscript.

Reviewer 2 Report
The manuscript is well-written (mainly), reports about interesting original results and, surely, can be accepted after some improvements. In short, authors demonstrate good expertise and experience for in-situ Raman and the following PCA analysis, but have not demonstrated the same for LFP as an electrode material.
1. I cannot agree with the manuscript title. “Full scenario …” requires much deeper insight. Please use more accurate wording.
2. I would recommend adding at least one more Figure from Supplementary materials (Figures S1 and/or S2). Authors can easily join Figure 2 and 3 – the journal one-column style makes it reasonable due to article design optimization.
3. Introduction can be improved by adding one paragraph about Raman spectroscopy in LFP or, at least, in-situ Raman spectroscopy in LFP.
4. Short description of PCA method advantages/peculiarities will be helpful.
5. Characteristic Raman bands of the electrolyte are not mentioned. Please reason why electrolyte does not interfere LFP Raman spectra.
6. The motivation of using spectral range only above 900 cm-1 is not provided in the beginning (starting from line 77). The usage of spectral range from 930 to 980 cm-1 (and above for some manuscript parts) is given too late (lines 89-90).
7. Lines 78 and 80 – please add citation referring to characteristic band positions for FePO4 and LiFePO4 correspondingly.
8. Line 83-84 – you measure Raman spectra at one point, but speak about sample/material in general. It would be more correct to discuss or, at least, mention the method locality.
9. Line 95 – “This difference means that …” – please use more accurate statement
10. Line 99 – please change the word “performance”. I’m not sure that you can conclude about performance and, as well, you use word “performed” lately in the same sentence.
11. Line 208 – “Phase transition mechanism can be determined …” – again please use more accurate and informative/detailed statement.
Author Response
All the replied are written in boldface.
We appreciate reviewer’s comments on our manuscript which have made us to revise the manuscript significantly.
Response on comments of Reviewers #2
The manuscript is well-written (mainly), reports about interesting original results and, surely, can be accepted after some improvements. In short, authors demonstrate good expertise and experience for in-situ Raman and the following PCA analysis, but have not demonstrated the same for LFP as an electrode material.
1. I cannot agree with the manuscript title. “Full scenario …” requires much deeper insight. Please use more accurate wording.
According to Reviewer’s comments, we revised the manuscript title in the revised manuscript.
“Investigation of the phase transition mechanism in LiFePO4 cathode using in situ Raman spectroscopy and 2D correlation spectroscopy during initial cycle”
2. I would recommend adding at least one more Figure from Supplementary materials (Figures S1 and/or S2). Authors can easily join Figure 2 and 3 – the journal one-column style makes it reasonable due to article design optimization.
According to Reviewer’s comments, we added Figure S1 as Figure 1 and revised all Figure number accordingly in the revised manuscript.
3. Introduction can be improved by adding one paragraph about Raman spectroscopy in LFP or, at least, in-situ Raman spectroscopy in LFP.
According to Reviewer’s comments, we added the sentence in the revised manuscript.
“The phase transition mechanism of LiFePO4 has been also investigated through in situ Raman spectroscopy[4,13].”
This content appears at the Introduction (in page 2) of the revised manuscript.
4. Short description of PCA method advantages/peculiarities will be helpful.
According to Reviewer’s comments, we added description of the PCA method in the revised manuscript.
“Principal component analysis (PCA) is a very popular analytical technique for spectroscopy. It can decompose the original data into the products of scores and loading vectors and provides a precise a precise mathematical estimation of changes along sample and variable vectors. From PCA results, we can capture the dynamics of systems in response to perturbations and to display relationships among variables or among variables and systems [42-44].”
This content appears at the Introduction (in page 2) of the revised manuscript.
5. Characteristic Raman bands of the electrolyte are not mentioned. Please reason why electrolyte does not interfere LFP Raman spectra.
In this study, we only focused on the phase transition between LiFePO4 and FePO4 phases without electrolyte region in Raman spectra.
6. The motivation of using spectral range only above 900 cm-1 is not provided in the beginning (starting from line 77). The usage of spectral range from 930 to 980 cm-1 (and above for some manuscript parts) is given too late (lines 89-90).
In this study, we only focused on the phase transition between LiFePO4 and FePO4 phases without electrolyte region in Raman spectra. The carbon band region (1150-1800 cm-1) was used normalization of the Raman spectra. Therefore, we used only above 900 cm-1 spectral range in this study.
7. Lines 78 and 80 – please add citation referring to characteristic band positions for FePO4 and LiFePO4 correspondingly.
According to Reviewer’s comments, we cited reference [13] in the revised manuscript.
“The bands at 960 and 1020-1130 cm-1, assigned to symmetric and asymmetric stretching of PO43- in FePO4 phase[13], ….”
This content appears at the Results and Discussion (in page 2) of the revised manuscript.
8. Line 83-84 – you measure Raman spectra at one point, but speak about sample/material in general. It would be more correct to discuss or, at least, mention the method locality.
According to Reviewer’s comments, we added spot size of the laser in the revised manuscript.
“The power and spot size of the laser were about 12.5 mW and 8 mm, respectively.”
This content appears at the Materials and Methods (in page 6) of the revised manuscript.
9. Line 95 – “This difference means that …” – please use more accurate statement
According to Reviewer’s comments, we revised the sentence in the revised manuscript.
“This difference clearly indicates that ….”
This content appears at the Results and Discussion (in page 3) of the revised manuscript.
10. Line 99 – please change the word “performance”. I’m not sure that you can conclude about performance and, as well, you use word “performed” lately in the same sentence.
According to Reviewer’s comments, we changed “performed” to “carried out” in the revised manuscript.
This content appears at the Results and Discussion (in page 3) of the revised manuscript.
11. Line 208 – “Phase transition mechanism can be determined …” – again please use more accurate and informative/detailed statement.
According to Reviewer’s comments, we revised the sentence in the revised manuscript.
“Phase transition mechanism of the LiFePO4 cathode in LiFePO4/Li ion cell can be examined using 2D correlation spectra obtained from in situ Raman spectra during the first and second charging-discharging processes.”
This content appears at the Conclusions (in page 7) of the revised manuscript.
